# On the Use of Phylogeographic Inference to Infer the Dispersal History of Rabies Virus: A Review Study

**DOI:** 10.3390/v13081628

**Published:** 2021-08-17

**Authors:** Kanika D. Nahata, Nena Bollen, Mandev S. Gill, Maylis Layan, Hervé Bourhy, Simon Dellicour, Guy Baele

**Affiliations:** 1Department of Microbiology, Immunology and Transplantation, Rega Institute KU Leuven, 3000 Leuven, Belgium; nena.bollen@kuleuven.be (N.B.); mandev.gill@kuleuven.be (M.S.G.); simon.dellicour@ulb.ac.be (S.D.); guy.baele@kuleuven.be (G.B.); 2Mathematical Modelling of Infectious Diseases Unit, Institut Pasteur, Sorbonne Université, UMR2000, CNRS, 75015 Paris, France; maylis.layan@pasteur.fr; 3Lyssavirus Epidemiology and Neuropathology Unit, Institut Pasteur, 75015 Paris, France; herve.bourhy@pasteur.fr; 4WHO Collaborating Centre for Reference and Research on Rabies, Institut Pasteur, 75015 Paris, France; 5Spatial Epidemiology Lab (SpELL), Université Libre de Bruxelles, 1050 Bruxelles, Belgium

**Keywords:** rabies, discrete phylogeography, continuous phylogeography, Bayesian inference, viral spread, environmental factors, pathogen phylodynamics, RABV

## Abstract

Rabies is a neglected zoonotic disease which is caused by negative strand RNA-viruses belonging to the genus *Lyssavirus*. Within this genus, rabies viruses circulate in a diverse set of mammalian reservoir hosts, is present worldwide, and is almost always fatal in non-vaccinated humans. Approximately 59,000 people are still estimated to die from rabies each year, leading to a global initiative to work towards the goal of zero human deaths from dog-mediated rabies by 2030, requiring scientific efforts from different research fields. The past decade has seen a much increased use of phylogeographic and phylodynamic analyses to study the evolution and spread of rabies virus. We here review published studies in these research areas, making a distinction between the geographic resolution associated with the available sequence data. We pay special attention to environmental factors that these studies found to be relevant to the spread of rabies virus. Importantly, we highlight a knowledge gap in terms of applying these methods when all required data were available but not fully exploited. We conclude with an overview of recent methodological developments that have yet to be applied in phylogeographic and phylodynamic analyses of rabies virus.

## 1. Introduction

Rabies virus (RABV) is the etiological cause of rabies, a fatal neurological infection in humans and other mammals, transmitted through the saliva of rabid animals via a bite or scratch [1]. RABV circulation is maintained through terrestrial and aerial cycles, associated with different species within the orders Carnivora and Chiroptera [2]. Vaccines to prevent rabies in humans have been available for over 100 years, and hence most deaths from rabies occur in areas with inadequate public health resources and limited access to such preventive treatments. Identifying current and potentially future at-risk communities and the factors that increase such risks are critical steps in combating RABV-related deaths [3].

Combating viral spread and the associated disease burden is a tremendous challenge requiring sustained research effort, to which viral sequence data represent a major asset. Inference of viral transmission dynamics from genetic data is typically based on concepts from phylogenetics and population genetics, but also links pathogen evolution to the dynamics of infection and transmission. Reconstruction of the unobserved—and typically time-stamped—phylogeny relating a molecular sequence sample thus serves as molecular epidemiologists’ primary tool [4], allowing them to tackle key biological questions on viral epidemics.

Integrating genetic data along with environmental factors and population dynamics within phylogeographic frameworks offers the opportunity to quantify how different attributes influence the transmission processes that underpin the circulation of rabies across and within countries [5,6]. These models can help us better understand RABV spread on a landscape level, and have been recognized as a promising tool in the fight against rabies [7,8]. Furthermore, with recent advances in high-throughput molecular sequencing, genomic data from pathogens are becoming available in unprecedented quantities and with remarkable speed, even in resource-limited settings, aided by portable genome sequencing technology [9]. Initial landscape phylogeographic studies such as the one by Biek et al. [10] along with the increasing availability of genomic data and the growth in computer processing power, paved the way for the development of complex statistical methods. This contributed to the current popularity of Bayesian phylogeographic and phylodynamic inference in infectious disease research [4]. Phylogeographic models extend existing evolutionary models by including metadata, such as the sampling location of the host species associated with a given sequence, and have led to a wide range of studies on the evolution and spread of pathogens [11].

Two different Bayesian inference methods are widely used in order to perform phylogeographic analyses, i.e., a discrete phylogeography approach [12] that uses discrete locations and a continuous phylogeography approach [13] for when fine-grained geographic data (e.g., latitude and longitude) are available. The latter type of information is often available in the case of RABV studies. However, if only the general location is known or the study spans multiple countries, a discrete phylogeographic model is typically used. Both of these approaches have had a strong impact on reconstructing viral spread and are incorporated in the widely-used BEAST software package [14].

Now that these approaches have been in use for over ten years, we compare both methodologies (see illustration in Figure 2) with a view towards answering the following questions: to what extent have these approaches been used to study RABV in different host species, in different parts of the world? What are the different types of conclusions that are reached using discrete phylogeography or continuous phylogeography? What are the advantages and disadvantages of each approach? We also focus on extensions of these phylogeographic models that allow for the use of covariate data by, for example, using generalized linear models. Factors such as host density, landscape features and geopolitical borders have been widely studied to better understand the spatial spread of the RABV. We review these factors along with several others that are typically analysed in RABV studies. Finally, we also discuss recent methodological advances that offer alternative phylogeographic inference frameworks. We note that this is not an exhaustive review of the literature. Some important rabies-endemic regions (most notably India) were not (yet) studied using these phylogeographic inference methods, and the studies presented in this review hence do not offer a complete overview of the dispersal history of RABV lineages across the world. Instead, our goal is to highlight the insights generated from the application of these methods, to synthesise the state of the art and to outline novel approaches developed to study pathogen phylodynamics. We refer to a recent scoping review [15] for a detailed overview of mathematical modelling of disease dynamics and phylodynamics to characterize dog rabies dynamics and control.

## 2. (Lack of) Genomic Data, Geographic Scope and Methodological Expertise

Genomic data can provide important and unique insights into rabies spread and persistence that can inform control efforts [16]. High-throughput sequencing technologies can be used for rapidly obtaining the whole virus genome, which may offer increased discriminatory capacity and allow for a more targeted infection control response [17]. Notably whole-genome sequencing of viruses is a powerful tool for studying virus evolution and tracking outbreaks [18]. Despite their great potential to provide a better understanding of the processes determining rabies spread and persistence, few phylogeographic studies on RABV have made use of whole genomes (see Tables 1 and 2).

Genomic data for rabies are used infrequently, and their use is usually limited to research studies that perform retrospective analyses that may not have an actionable impact in real time [16]. Brunker et al. [16] provide a framework for shortening sequencing time to two to three days in an affordable manner, sometimes even reducing costs by about half the price. This is particularly useful for low and middle-income countries where genomic research is challenging due to insufficient infrastructure, shortage in supply chain or logistical obstacles. Brunker et al. [16] demonstrate the feasibility of real-time sequencing of RABV to rapidly inform policy decisions and disease management across different locations in Kenya, Tanzania, and the Philippines.

The availability of data is a considerable challenge in effectively studying RABV dispersal dynamics, and is closely tied to its geographic scope. Both of these factors typically influence the phylogeographic methods that researchers choose to employ. Compared to the studies using discrete methods, for instance, the geographic areas under consideration in continuous phylogeographic studies are generally much smaller. Furthermore, discrete phylogeographic methods are often employed in preliminary analyses before conducting continuous phylogeographic analyses [19,20]. The smaller geographic scope of continuous phylogeographic analyses can be partly explained as a data-availability issue, given that a continuous approach requires precise geographic coordinates. Rabies diagnosis requires a brain biopsy and it is thus difficult to diagnose antemortem [21]. Most of the time, rabies samples are obtained from roadkill or euthanized suspected rabid animals, and available data are hence sporadic in nature. Reliably tracking the precise movement of rabid animals across vast geographic areas and across political borders depends on sustained surveillance efforts, and would require considerable coordination and cooperation [16,22,23]. The ARTIC network, a Wellcome Trust-funded project to develop the application of genomic surveillance for viral outbreak response, provides comprehensive open-source resources for laboratory and sequencing work, bioinformatics, phylogenetics and subsequent data analyses [16].

In our literature review, we encountered several epidemiological studies in which the sequence data were accompanied by either precise geographic coordinates or discrete locations, but no phylogeographic methodologies were applied [24,25,26,27,28,29,30,31,32]. In many cases, exploring the geographic spread of RABV was explicitly one of the study goals. Two of these papers even employed BEAST [14] in order to construct phylogenetic trees, but stopped short of conducting phylogeographic reconstructions. We aim to achieve a more widespread awareness of the available phylogeographic methodologies, and particularly the insights that they can provide, in the hope that this leads to an increased usage and in turn to a better understanding of the worldwide spread of RABV. Towards the end of this study, we provide useful recommendations to perform these types of analyses, including possible approaches to deal with missing (genomic and location) data. We also found that, despite precise geographic coordinates being available, the application of phylogeographic methods was not possible due to missing sequencing data [33,34,35]. This is especially an issue for research in developing countries, where sequencing efforts are often hindered by their expense. For example, while costs are estimated to have dropped to GBP 60 per sequence, and are estimated to drop further as workflows are optimised [16], this can still pose a significant hurdle in developing countries, where R&D expenditure often accounts for less than 0.5% of the GDP.

A lack of funding is only part of a broader problem: one of unequal access to scientific advances and the impact of this inequality on the fight against rabies. For example, several surveys in developing countries reported a deficiency of knowledge on how to treat wounds from dog bites [23,36,37]. We refer to Figure 1 as an illustration. Many countries stand out as being either over-represented or under-represented in terms of sequencing efforts: developed countries have not seen a human death due to rabies in years, but they still produce a non-negligible number of sequences (sampled from non-human hosts). Meanwhile, in Central Africa and Southeast Asia few sequences are produced compared to the burden of rabies. A striking example is India, which accounts for more than 50% of human rabies deaths worldwide [38], but for which we found no phylogeographic studies.

## 3. Discrete Phylogeographic Inference

Genomic data now often have associated discrete locations of sampling, in the form of a municipality, district, province, or country. Examining the (location) traits associated with sequence data in an evolutionary context requires a model of how these traits evolve throughout evolutionary history [11]. In such a case, one can posit a continuous-time Markov chain (CTMC) model that begins at the root of the phylogeny and proceeds down the phylogeny to its tips (which are associated with the observed data), acting independently along the branches. Here, the CTMC state space consists of discretized sequence sampling locations. This is analogous to the standard CTMC-based modelling of molecular sequence character evolution along phylogenies [39]. While this discrete CTMC-based model allows for movement between any two sampling locations in the spatial reconstruction, such movements are very unlikely for most pairs of geographic locations under consideration. The corresponding transition rates in the model that are expected to be zero can lead to extremely high variance estimates for the inferred ancestral locations. Fortunately, this limited data problem can be overcome by a parsimonious selection of parameters using Bayesian stochastic search variable selection (BSSVS) [12]. BSSVS, which is traditionally applied to model selection problems in a linear regression framework, enables us to simultaneously determine which transition rates are zero depending on the evidence in the data and efficiently infer the ancestral locations [12].

In addition to reconstructing the spatio-temporal spread of a virus, one can examine the relationship of covariates with this spread. This is done by extending the discrete phylogeographic model using a generalised linear model (GLM) for the transition rates between locations. This approach, developed by Lemey et al. [40], parameterizes the transition rate from one location to another as a log-linear function of several potential covariates. Importantly, it is necessary to have covariate data for all discrete locations in the model. Again invoking BSSVS, various combinations of covariates can be tested so that we obtain both the probability of inclusion of a certain covariate as well as its effect size. This GLM approach has been employed, for example, to analyse the effect of the global air transportation network on seasonal influenza [40].

Table 1 shows an extensive overview of studies that reconstructed the spatial dispersal of RABV using discrete phylogeographic inference. As previously mentioned, discrete phylogeographic analyses are sometimes performed as preliminary analyses before performing continuous phylogeographic analyses. Such an approach may be used to identify the different RABV clades in circulation and the potential introductions of the virus into potential areas of interest.

Table A1 shows those studies that employ discrete phylogeographic inference in combination with collected covariate data that may act as relevant predictors of RABV spread. Some of those covariate data can however be hard to come by and can be part of the reason why geographical features, along with their clear impact on the habitats and movement of species carrying RABV, are predominantly used in these types of analyses.

### 3.1. Host Species

As can be seen from Table 1, the past twelve to fifteen years have seen a wide range of studies on the spread of RABV in different host species. Most cases of human rabies result from dog bites in developing areas where canine rabies is common, whereas rabid wild animals are usually responsible for human rabies in regions where dogs are vaccinated [59]. There are several different species that have been reported to spread rabies in different parts of the world, including raccoons and skunks in North America, bats and dogs in South America, dogs in Asia, and badgers in Taiwan. We here delve into more detail regarding those studies and focus on the various carrier species encountered in our literature search.

#### 3.1.1. Dogs

Carnieli et al. [48] found that the lineages in the southeastern region of Brazil in the 1970s were closer to the most recent common ancestor of all the lineages than the lineages in the midwestern, northern and northeastern regions. Based on previous studies [60,61,62], Carnieli et al. [48] hypothesize that the urbanization of new regions, development of roads, human migration have a major impact on the spread of rabies. Carnieli et al. [48] suggest that the move of capital to the midwestern region of Brazil experienced significant economic growth along with migratory flow of humans causing the spread of RABV from southeastern region to other regions in Brazil. Testing socio-economic factors within a GLM framework will provide further details about the spread of RABV on a regional level.

The role of human activities in mediating the spread of dog RABV has been continually examined using phylogeographic methods over the past decade, with studies incorporating a variety of factors ranging from natural geographical barriers to human population density. Talbi et al. [43] used a GLM to show that road distances are better predictors of the spatial spread of dog RABV in northern Africa than spatial accessibility or raw geographical distance. The authors observed occasional long-distance (>200 km) dispersal within a span of 1–2 years, and that the RABV diffusion process was restricted by geopolitical boundaries at larger scales. As this is inconsistent with what was observed in endemic wildlife rabies [10], it is clear that humans have played an active role in the spread of dog RABV. Similar evidence of long-distance migration and a high dispersal rate has been found in Tanzania [41]. Using GLMs, Brunker et al. [6] found that the presence of dogs, but not their density, was of importance in the spread of rabies in the Serengeti district in Tanzania. The authors also found that while rivers acted as barriers, roads acted as facilitators of the spread on a large scale.

Hayman et al. [42] studied the spread of RABV in dogs in Ghana. They believed that initially the large tropical forest system along the Ghana-Ivory Coast border provided a barrier to dog movements and hence fewer viruses were from the Ivory Coast. Rapid deforestation and increasingly easy inter-country travel were put forth by the authors as the reasons behind the trans-boundary movements of RABV. As non-human involvement would require viruses to be transmitted at an approximate rate of between 39 to 53 km per year, the study showed that the more likely reason for this virus’ presence in Ghana was that an infected animal was trans-located from the east, thus introducing a new sub-lineage to the region. RABV epidemiology was more intricate than expected in West Africa and there were repeated introductions of RABV into Ghana [42]. This analysis also highlighted the potential problems of independent developing countries implementing rabies control programs in the absence of a regional one asking for greater cooperation in combating rabies.

Omodo et al. [23] observed mixed lineages circulating between northern and western Uganda which could have been associated with the movement of dogs or wild animals such as foxes, jackals, mongoose and hyenas along these regions (Lake Albert and the Semliki River). The cross-border movements of animals between Sudan, Democratic Republic of Congo, Tanzania, and Uganda could potentially have contributed to the introduction of two lineages in Uganda seen in the study [23]. The authors stated that the location of Uganda at the crossroads of three major biogeographical regions (Ethiopio-Somalian, Sudano-Congolian, Zambezian) may have favored the local circulation of different rabies lineages from adjacent regions. Phylogenetic analyses in the study linked the circulation of one lineage in Uganda (detected from 2010 to 2012 in all localities sampled) to a Tanzanian lineage and the episodic presence of a second lineage (in 2010 in Moyo), also to a Tanzanian lineage.

For a human rabies case in South Africa, the use of phylogeographic methods was crucial to understand transmission pathways and incidences of long-range transmission and introductions from distant or separately administered regions [44]. Specifically, it lent confidence to the assessment that the sporadic human rabies case represented neither a failure of surveillance nor the re-emergence of rabies in a carefully maintained rabies-free area, and showed how other countries could potentially benefit from effective surveillance.

Inter-island transportation of infected animals, mainly dogs, is one of the main reason behind the persistence of rabies in Indonesia [51]. Using a species-annotated tree inferred through discrete trait methods, Dibia et al. [51] revealed that the infections in other animal species (cats, goats and pigs) originated from dogs. Through a phylogeographic analysis, they also revealed that the risk remains high for newly rabies-free areas and intensified control areas such as Bali. Philippine RABV strains were introduced from China around the beginning of the 20th century. Tohma et al. [50] found that upon this introduction, the RABVs evolved within the Philippines to form three major clades, and there was no indication of introduction of other RABVs from any other country. The phylogeographic reconstruction in the same study [50] revealed island-to-island migrations within the Philippines. Importantly, it showed that the evolutionary pattern of these viruses was strongly shaped by geographical boundaries, indicating that the seas were a significant geographical barrier for viral dispersal.

Guo et al. [55] indicated that within Southeast Asia, isolates mainly clustered according to their geographic origin. They found evidence of sporadic exchange of strains between neighboring countries, but it was shown that the major strain responsible for the Chinese epidemic in the 2010s (i.e., during the third rabies epidemic in China since 1949; Tao et al. [24]) had not been exported. They claimed that the national geographical boundaries and border controls were effective at halting the spread of rabies from China into adjacent regions. They also found that the epidemic in 2013 was dominated by variant strains that were likely present at low levels in previous epidemics in China. Guo et al. [55] affirmed the accuracy of the phylogeographic analyses with epidemiological linkages between high-incidence provinces consistent with observations based on surveillance data from human rabies cases.

Wang et al. [57] detected six sub-clades of RABV circulating in China and found that each of them had a specific geographical distribution, reflecting possible physical barriers to gene flow. Their phylogeographic analysis revealed minimal viral movement among different geographical locations. Zhang et al. [56] used phylogeographic analyses to reveal that in China, within-country circulation accounted for more infections than virus importation. The authors observed that the entry of RABV into southern China from Southeast Asia was the only well-supported case of virus importation into China.

Ma et al. [37] detected viral migration paths from Sichuan, Guizhou, and Hunan to the Hanzhong prefecture of Shaanxi, followed by viral spread to Xi’an and other prefectures. As only Sichuan is adjacent to Hanzhong prefecture of Shaanxi province, the rabies strain in Shaanxi might have come from the neighboring Sichuan province [37]. The range of rabies-affected areas has extended in China, spreading from the east and south to the west and north as noted by Ma et al. [37]. They suggest that this pattern may be the result of the gradual development of a transportation network which increased the opportunity for rabid dogs to move or be transported among different areas. It would be interesting to test this hypothesis formally by including road distances as a variable in a GLM framework.

Horton et al. [49] found no evidence that the RABV strains circulating in Iran were direct descendants of virus ancestors that existed in the region 4000 years ago. In contrast, these analyses supported at least one introduction of rabies from Europe with subsequent spread, albeit on a markedly different timescale, in the last 150 years. They reported co-occurrence of distinct lineages in Iran which was also seen in an analysis by Dellicour et al. [20]. Horton et al. [49] suggested that these lineages corresponded to independent introductions of rabies in Iran, highlighting the importance of the geographical position of the Iranian region. With the exception of two viruses detected in Iran from the Arctic-like lineage, the study found an apparent barrier to the spread of the Arctic-like lineage at approximately 60 degrees longitude, corresponding to the Iranian border. The study also found that there is a clear and strongly supported distinction between the viruses circulating in Pakistan and Afghanistan, and those further west. In contrast with the Iranian border as a barrier to eastward spread, there was significant support in the study for the spread of rabies among countries to the west of Iran. Their evidence of descendants of a wildlife-associated lineage in dogs in Turkey and Azerbaijan revealed that wildlife and dog rabies were not as distinct.

#### 3.1.2. Bats

Phylogeographic analyses have provided statistical support for at least three independent introductions of RABV into Trinidad from the mainland, favoring Brazil and Uruguay as source populations [45]. The analyses show three largely temporally defined lineages within the phylogeny from which each of the three Trinidadian lineages arose. The reconstructions in the study revealed that the lineages belonged to a widespread clade of RABV variants perpetuated by *Desmodus rotundus* (vampire) bats. This suggests that the Trinidad outbreaks most likely originated from rabid *Desmodus rotundus* bats. The study showed the dates of divergence from the Brazilian and Uruguayan ancestors predated lineage expansion within Trinidad by several years. Seetahal et al. [45] suggested that the spread of RABV by *Desmodus rotundus* bats occurred via gradual movement of infected bats flying to regions of the mainland neighbouring Trinidad with subsequent entry. They believed that during the years before the expansion of lineages in Trinidad, RABV from Brazil and Uruguay spread northward from country to country on the mainland until being isolated in Trinidad several years later. This migratory pattern of vampire bats would be interesting to explore using continuous phylogeographic inference as such an approach would infer a spatial velocity for RABV in this area, and certain landscape features could be tested as factors that facilitate or impede movement.

Streicker et al. [47] conducted a discrete phylogeographic analysis to identify RABV clades in Costa Rica and inferred five distinct clades with origins in North and South America. Their study indicated bidirectional viral dispersal involving countries to the north and south of Costa Rica at different time points.

#### 3.1.3. Raccoons

Trewby et al. [46] observed that in Canada, in and near the province of Quebec, vaccinated areas are still vulnerable to long-distance translocation events, effectively allowing RABV infection to bypass areas of vaccination completely. The authors stated that if such events were the result of local spreading, the sampled sequences would have been genetically similar to the sampled sequences from the US-Canada border. However, their phylogenetic analyses showed that this was not the case and hence the outbreak was a result of either a long-distance transmission or non-exhaustive sampling. These phylogeographic analyses strongly supported the backflow of infection from Quebec across the border into the US. The analyses also revealed that some areas experienced multiple incursions, either in short succession (Ontario, Quebec) or separated by several years (New Brunswick) [46]. The authors suggested the lack of natural barriers along the Quebec border to be a likely contributing factor for the higher transboundary transmission, compared to the major rivers or lakes that reinforce the border between Ontario-US. These deterministic factors can be tested in a follow-up analysis using a GLM framework. Other potential contributing factors that affect the temporal and spatial variation in raccoon demography or vaccination coverage affecting local pressure of infection as suggested by Trewby et al. [46] can also be tested using GLMs.

#### 3.1.4. Badgers

Lin et al. [52] found that the transmission of RABV in ferret badgers likely originated from Eastern Taiwan, then moved across the central region to western regions. Northern Taiwan however, remained a rabies-free zone despite the existence of ferret badgers and exhaustive testing [63,64]. Lin et al. [52] suggest that the Da-An River was a natural barrier that prevented the spread of RABV to the northern region. However, the study states that the reason for the river to be a barrier is unclear. As the Da-An river was not examined using a formal statistical test to be a barrier for RABV spread, valuable insights could be attained by incorporating the geographical barriers of Taiwan in a GLM framework.

According to the location-annotation phylogeny reconstructed by Lan et al. [53], three major genotypes of ferret badger RABV circulated in three different geographical areas in Taiwan. They observed that two genotypes had distributed into central and southern Taiwan between two ecological river barriers (the Da-An River and the JhuoShuei River), and the third genotype had been limited in southeastern Taiwan by the Central Mountain Range.

#### 3.1.5. Livestock

Yu et al. [54] suggested that the 2007 epidemic in China was primarily composed of a younger strain (1992) with a geographical dispersion that was consistent with the recorded spread of the virus, and an older strain (1960) that corresponded to a previous epidemic. Their analyses revealed that this latter group exhibited a different geographical pattern, and that this strain remained at low levels throughout the country and was able to re-emerge as the epidemic took hold. They discovered a small number of migration events played the major role in the spread of the virus. This was also supported by the observation that the branch order in the tree coincided with epidemiology data that showed that the neighboring provinces of Hunan, Guangxi and Guizhou experienced rabies outbreaks sequentially. The authors also showed that Hunan in southwest China served as a major source of geographic dispersal. Additionally, they identified locations such as Jiangsu that acted as popular migration event spot and aided dissemination of the virus. As the reasons why Jiangsu acted as a major migration source was unclear, using socio-economic variables including trade data in a GLM could reveal underlying factors that impacted the spread of the virus. The phylogenetic analyses of Yu et al. [54] placed ferret badger sequences at the top of two distinct sub-clades of samples isolated from dogs, demonstrating that the rabies in wildlife was not a consequence of spillover from dogs.

Yu et al. [58] showed that the geographic origin of the Arctic-like (AL) RABVs was in Siberia and the far-east region in Russia approximately in 1830s. The authors found that the ancestral AL RABV then diversified and immigrated from east Russia to countries in Northeast Asia, while the viruses in South Asia were dispersed to the neighboring regions from India. It is important to note that RABV sequences from the Indian subcontinent were not included in this study. Yu et al. [58] report that the migratory event between east Russia and India was not statistically supported, as assessed through its corresponding Bayes factor. Therefore, the migration trajectory of AL lineages between east Russia and India remains uncertain. Dogs, domestic animals and raccoon dogs accounted for the overwhelming majority of the distribution in the reconstructed ancestral host. Yu et al. [58] observed the dispersal of AL RABVs from South Asia to West Asia via two transmission routes: one was the India-Nepal-Iran route, and the other involved transmission along the India-Pakistan-Afghanistan-Iraq route. The reconstruction revealed that after the 1990s, the South Asian countries, especially India, witnessed a large number of AL RABVs cases and the viruses rapidly spread to Nepal, Bangladesh and Bhutan. The dispersal between India and Nepal, India and Bangladesh, and Bangladesh and Nepal, and between Bangladesh and Bhutan was statistically supported by high Bayes factors. They however observed no viral exchange between the South Asian countries and West China (Qinghai and Tibet provinces). We suspect that the natural barrier of a high-altitude Himalayan landscape between the two regions halted the spread of the viruses, which can be formally tested using a GLM Figure 2.

## 4. Continuous Phylogeographic Inference

When (more) fine-grained geographic information about sequence sampling locations is available, it is possible to use a continuous phylogeographic inference to reconstruct viral spread over time. Lemey et al. [13] introduced a model that reconstructs the viral dispersal history in continuous space via a two-dimensional relaxed random walk. Here, the two dimensions correspond to latitude and longitude coordinates and the phylogeographic reconstruction is thus spatially-explicit. The relaxed random walk starts at the root of a phylogeny and proceeds down its branches, with the change in location from one end of a branch to the other being normally distributed. Importantly, this model accommodates heterogeneity in the viral dispersal by allowing the diffusion rate of the process to vary along the different branches of a phylogeny. This is crucial for realistic spatio-temporal reconstructions.

While a discrete phylogeographic approach has proven to be useful in a wide range of scenarios, it presents several disadvantages compared to the continuous approach. First, discrete phylogeography requires an arbitrary grouping of geographic locations that can be an unrealistic, oversimplified representation of the area of study. Furthermore, the range of geographic locations is restricted to the sampling locations, and requiring the locations of all ancestors of the samples to correspond to a subset of the sampling locations can be unrealistic. Finally, discrete phylogeographic reconstruction is highly susceptible to sampling bias. Undersampling or oversampling from certain locations can substantially impact the estimates of transition rates between locations and, in turn, spatial reconstructions [65]. A disadvantage of the continuous approach is the need for the geographical information to be as precise as possible, with less recent data sets only having more coarse-grained location information (e.g., province or country of sampling) available.

Table 2 shows an extensive overview of studies that reconstructed the spatial dispersal of RABV using continuous phylogeographic inference. The requirement of having precise sampling coordinates has resulted in fewer such studies being available as compared to those performed using discrete phylogeographic inference, as listed in Table 1.

As with the discrete phylogeographic framework, one can examine the relationship of covariates with viral spread. In a continuous phylogeographic inference setup, testing the association of environmental factors with viral spread is however currently performed in a post-hoc manner. Such an analysis is hence split up into two steps: a first step performing continuous phylogeographic inference to generate a set of location-annotated phylogenetic trees, followed by a second step to assess the impact of environmental factors on viral spread (in particular on the dispersal velocity of viral lineages), conditional on the collected phylogenies [66]. This assessment makes use of an environmental raster to compute environmental distances (or “weights”) for each phylogeny branch, which represent the degree to which the environmental variable impedes (or facilitates) lineage movement. Correlations between movement duration and these environmental weights are then assessed, and the statistical significance of these correlations are evaluated using null distributions generated by a randomization procedure.

### 4.1. Host Species

As can be seen from Table 2, the past ten years have also seen quite a range of studies on the spread of RABV in different host species. Table A2 shows those studies that perform a post-hoc inference in order to test the impact of environmental factors on viral spread. Interestingly, our list of publications shows that these studies have to a lesser extent focused on RABV epidemics in Asia and to a larger extent on studying the situation in Latin America. We here again delve into more detail regarding these continuous phylogeographic studies and focus on the various carrier species analysed in those studies.

#### 4.1.1. Dogs

Carnieli et al. [69] analysed a data set of RABV isolates from North and Northeast Brazil with the aid of continuous phylogeographic methods. The authors inferred that the most recent common ancestor of the samples became established at the end of the nineteenth century on the border of the Brazilian states of Paraíba and Pernambuco and diversified into the lineages associated with dogs and crab-eating foxes (*Cerdocyon thous*). Carnieli et al. [69] found that around 1910, the original *C. thous* lineage diversified into two main sub-lineages in the same area while the dog-associated lineage diversified around 1945 and moved towards the north and south. The authors deduced that the dog-associated lineages dispersed at an average rate of 30.5 km/year and the *C. thous*-associated lineages dispersed at an average rate of 9.5 km/year. The authors showed that the dispersion of RABV lineages isolated from dogs followed human activities and was associated with urban centers. The dispersion of lineages isolated from *C. thous*, on the other hand, was shown to reflect this animal’s solitary habits and ecological niches. The results of this study indicated that the genetic identities of the two RABV sub-lineages isolated from *C. thous* were maintained because one of them spread to the southern part of the Northeast Region of Brazil while the other moved to the north of the same region, suggesting that *C. thous* sub-populations found in different areas acted as hosts for and transmitted each of the two sub-lineages intraspecifically. The phylogeographic analysis in this study made it possible to infer not only the movement of the virus lineages but also the probable location where dispersion and diversification occurred, proving it to be useful for reconstruction and surveillance [69].

The use of discrete phylogeographic methods by Dellicour et al. [20] revealed at least eight RABV clades in Iran, each with independent introductions of rabies in Iran. The authors’ analysis of metadata associated with these clades suggested relatively frequent RABV transmissions between dog and wildlife animal populations, and vice versa. Similar complex transmission patterns were identified in Tanzania [6] as well as in Turkey [49]. The continuous phylogeographic analysis by Dellicour et al. [20] in Iran highlighted that viral lineages tended to spread towards and remain in accessible areas associated with relatively high human population density. In addition, their analysis underlined that lineages were less likely to spread towards grasslands and to occur in barren vegetation areas. They also suggested that populated areas represent strategic places for vaccination campaigns because they can act as crossroads of transmission chains.

#### 4.1.2. Bats

Vampire bats are major rabies reservoir hosts and have affected the cattle population across the globe [2]. Torres et al. [2] showed that in highly populated areas in Argentina, vampire bats fed almost exclusively on cattle. It was previously observed by Delpietro et al. [73] that the bat population reached higher densities in human-populated areas and lived in roosts mainly located in human buildings and could inhabit areas that lacked natural roosts if prey were available [74]. Continuous phylogeographic inference of vampire bat RABV in Argentina by Torres et al. [2] revealed that a slower dispersion in the northwestern region correlated with the presence of geographical barriers that made raising livestock more difficult, while a faster dispersion in the northeastern region occurred due to the presence of uninterrupted grasslands favouring denser livestock areas. The authors concluded that RABV transmission dynamics in Argentina were characterized by initial epizootic waves followed by local enzootic cycles with variable persistence, with multiple foreign introductions possibly from Brazil.

In Brazil, Vieira et al. [70] found that RABV followed a centrifugal dispersion pattern with a mountain barrier being the only resistance factor curbing spread. While studying vampire bats RABV in Peru, Streicker et al. [71] found that male bats spread the virus between genetically isolated female populations. The study further indicated unanticipated gene flow through the Andes mountains connecting the RABV-free Pacific coast to the RABV-endemic Amazon rainforest. Using Bayesian phylogeography with landscape resistance models, Streicker et al. [71] projected invasion routes through northern Peru that were validated by real-time livestock rabies mortality data. Their results implied that while female vampire bats tended to stay in a particular area, male vampire bats likely contributed disproportionately to rabies spatial spread. Moreover, the authors suggest that setting up barriers to male dispersal will delimit the boundaries of viral distributions. The phylogeographic reconstructions from this study therefore provide useful insights that could help in curbing the spread of RABV due to bats.

Streicker et al. [47] found that within Costa Rica, viruses showed little contemporaneous spatial overlap and no lineage was detected across all years of surveillance (2004–2017). GLM tests suggested that lineage disappearances were more likely to be explained by viral extinctions than undetected viral circulation (which also explains why culling is not all that helpful). Their results suggested a Central American corridor of RABV invasions between North and South America, and showed that apparent disease endemicity may arise through recurrent pathogen extinctions and re-invasions which can be readily detected in relatively small data sets by joint phylogeographic inference (i.e., including a GLM).

#### 4.1.3. Skunks

Continuous phylogeographic analyses showed that rivers (with the exception of the Mississippi River and Rio Grande River) and roads did not constitute significant barriers for skunk RABV as compared to deserts or mountains [22,67]. The analyses by Kuzmina et al. [22] showed slow dispersal rates in skunk RABV as compared to raccoon or fox RABV. While reconstructing the spatial spread, the same study showed that skunk RABV exhibits a similar large-scale expansion at the present time compared to what was observed in the mid-Atlantic raccoon rabies epizootic [10,13]. This suggested strategic areas (such as the Mississippi River valley) for initiation of local or step-wise oral vaccination campaigns. The analysis of temporal dynamics by Pepin et al. [67] showed that skunk rabies were most likely to spread to new areas during the first half of the year, when skunk populations were producing new offspring.

#### 4.1.4. Raccoons

As the land in the Florida peninsula is homogeneous without significant barriers, the speed of raccoon RABV diffusion was also found to be spatially homogeneous by Musial et al. [68]. The emergence of strong phylogeographic structure in the virus was seen by the authors in the form of five monophyletic lineages that diverged during the early years of colonization and went on to each occupy a distinct sub-region of Florida. Based on samples taken over multiple decades, Musial et al. [68] showed that the spatial distribution of these lineages changed little since the 1970s. This phylogeographic stability allowed them to retrospectively identify a small set of counties within Florida as the likely source of the virus strain that seeded a much larger rabies outbreak in the northeastern USA in the 1970s. All of the viral clades detected in the study diverged while rabies expanded through a novel host system, suggesting that colonization and expansion processes drove the viral dispersal patterns. The study indicated that the mean center of each clade’s geographic distribution moved very little and the general locations of these centers were often preserved over multiple decades. This stability, as well as the absence of lineage turnover, implied that spatial genetic patterns in RABV were preserved through time, long after the initial invasion process. It also highlighted the overriding importance of local host movement processes, resulting in limited spatial admixture, in the maintenance of raccoon RABV.

#### 4.1.5. Foxes

Fox rabies re-emerged in northeastern Italy at the end of 2008 and circulated until early 2011. Fusaro et al. [72] identified two viral genetic groups referred to as Italy-1 and Italy-2. Phylogenetic and phylogeographic analyses in their study revealed that both groups had been circulating in the Western Balkans and Slovenia in previous years and were only later introduced into Italy and occupied different areas of the Italian territories. The authors showed that the RABVs belonging to the Italy-1 group remained confined to the region of introduction and their spread was minimised by the implementation of oral fox vaccination campaigns. On the other hand, they showed that Italy-2 viruses spread westward over a territory of 100 km from their first identification in the Friuli-Venezia Giulia region and likely crossed the northern territories where surveillance was inadequate. Fusaro et al. [72] noted the reduced passive surveillance in Italy in the period prior to the first rabies notification and the difficulty in retaining high awareness of field staff for rabies re-introduction in the areas that had experienced a rabies-free status for over ten years.

## 5. Meta-Analysis of Relevant Environmental Factors

While there have been studies explaining the dynamics of RABV in different hosts and habitats, Dellicour et al. [5] carried out a comparative meta-analysis of the dispersal dynamic of RABV lineages, re-analysing data sets involving various host species (raccoons, skunks, bats, and domestic dogs). The authors adapted the analytical procedure introduced by Dellicour et al. [75] to test the impact of miscellaneous environmental factors on the dispersal velocity of RABV lineages.

In their analysis of the skunk RABV data set, Dellicour et al. [5] found that human population density was associated as a supported conductance factor (i.e., a factor that facilitates movement). However, for the same data set, no supported association was found between lineage dispersal velocity and the ‘barren vegetation’ and ‘elevation’ environmental layers [5], which differs from a previous study of the same data set [22] that suggested deserts and mountains acted as barriers for viral spread. For the raccoon RABV data set, [5] confirmed that the elevation was supported a resistance factor (i.e., a factor that impedes movement), reaffirming what was previously highlighted by Biek et al. [10]. The analysis of a raccoon RABV data set by Dellicour et al. [5] also revealed that human geographic variables had a supported association with RABV lineage dispersal velocity: the ‘inaccessibility’ layer (grid of travel time to the nearest major city) and the ‘urban areas’ layer were identified as supported resistance and conductance factors, respectively. Dellicour et al. [5] further reiterated the influence of human activity in the high diffusivity of RABV in domestic dogs, similar to Talbi et al. [43] and Brunker et al. [41]: the three factors that seemed to impact RABV lineage dispersal velocity were mainly the presence of urban areas and human population density (identified as conductance factors), as well as ‘inaccessibility’ (identified as a resistance factor). For bat RABV data sets however, the meta-analysis showed that anthropological factors did not appear to have been associated with an important impact on the dispersal of those RABV lineages. In Latin America, the common vampire bat is a leading cause of human and animal rabies that result in unvaccinated livestock dying every year [70,71]. Owing to the complexity of RABV circulation in bat communities, none of the environmental factors tested by Dellicour et al. [5] proved to significantly impact the dispersal velocity of bat RABV lineages in their meta-analysis. One of the explanations put forward by the study was that the factors they tested may not have been the factors that are relevant to the ecology of bats. Sampling bias or sampling from a restricted area within a wider region of bat dispersal could also have compromised the statistical power necessary to identify relevant factors [5].

## 6. Recommendations and Useful Resources

In this review, we presented a wide range of applications to the study of RABV of two highly popular approaches to perform phylogeographic inference. In this section, we provide a few general guidelines on which of these approaches to employ for the different types of location data available, along with recent extensions of these approaches to deal with missing (genomic and location) data.

The key choice for which phylogeographic analysis to perform lies with the level of geographic precision for the location data associated with the collected sequences. Simply put, when very precise location data (e.g., GPS coordinates) are available for the collected sequences, continuous phylogeographic inference should be performed. If this is not the case, and only more coarse-grained location information (e.g., district, county, province or country), discrete phylogeographic inference can be performed. Both approaches have an associated framework to test the impact of environmental variables on viral spread, as discussed in this review, and this should hence not play a role in the decision of which approach to use. Additionally, both approaches lead to location-annotated phylogenetic trees as part of the end result, that can make for insightful visualisations. It is hence imperative that the collection of associated metadata, such as location data, be treated at the same level of importance as the collection of genomic data.

The decision to discard sequences because of missing associated location data is frequently made. Given the different types of data-availability problems associated with the study of RABV, we suggest to exploit recent developments to make use of as much genomic and location data as possible. For continuous phylogeographic analysis [13], missing coordinates can be dealt with by providing a polygon describing the known region of sampling [76,77]. If even this information is unknown, then the sequence will still have to be discarded. For discrete phylogeographic analysis [12], typically one sampling location is assigned to a sequence, although in the case of uncertainty multiple locations can be considered with equal probability (using ambiguity codes), as well as using specific probabilities based on different sources of information [78]. In the atypical case of having a known RABV infection with available location data but missing genomic data, the use of “ghost” or “sequence-free” samples can be considered [79,80].

Overall, for these different types of analyses, the http://beast.community/ last accessed on 10 August 2021 website provides a wealth of information in the form of tutorials that make use of example data sets that are also provided. A typical workflow for a discrete phylogeographic analysis on RABV in North American bat populations can be found on http://beast.community/workshop_discrete_diffusion last accessed on 10 August 2021, whereas two tutorials for continuous phylogeographic analysis are provided on http://beast.community/workshop_continuous_diffusion_yfv last accessed on 10 August 2021 and http://beast.community/workshop_continuous_diffusion_wnv last accessed on 10 August 2021, for yellow fever virus and West Nile virus respectively, although the exact same steps are used to set up a RABV analysis. As apparent from these webpages, different software packages are required to perform all of the required steps in the proposed tutorials. Download links for BEAST [14], BEAGLE (required to run BEAST; Ayres et al. [81]) and other related packages for processing and visualising the output of phylogeographic analyses can be found on http://beast.community/ last accessed on 10 August 2021 as well. While we discuss the application of phylogeographic methods on rabies virus (RABV) datasets here, it should be noted that these methods could also be used on genomic data from other zoonotic viruses, provided the data contain sufficient temporal signal. We elaborate more about the formal assessment of a temporal signal on available data in the next section.

## 7. Novel Methodological Developments and Future Perspectives

As was amply shown throughout this study, phylodynamic analyses for RABV are predominantly based on two types of phylogeographic models, i.e., one for discrete location data [12] and another for continuous location data [13], that were both developed over ten years ago and have an accompanying implementation in BEAST [14]. We refer to Baele et al. [11] for a thorough and more general overview of how to connect sequence evolution to trait evolution—where traits can comprise host, phenotypic and geographic sampling information—and how to incorporate covariates of evolutionary and epidemic processes in phylodynamic inference. Recent years have seen the development of additional methods to round out a typical phylodynamic workflow, as well as increased efforts to port these popular models into maximum-likelihood (ML) applications. With such a further adoption of these models also comes the need to assess the impact of sampling bias on phylodynamic inferences. While it is tempting to try to leverage these phylogeographic methods to predict (the locations of) future outbreaks through some form of forward simulation, this is far from straightforward as it entails drawing reliable patterns from relatively rare past events [82]. Additionally, the circumstances surrounding each outbreak are very dynamic, complicated and unique due to interactions between virus genetics, ecology and host factors [83]. This makes it extremely difficult to build a predictive model because such a model must be ‘trained’ on a very wide range of realistic scenarios in order to be reasonably accurate. Given such difficulties, it has been suggested that the most effective and realistic way to fight outbreaks is to monitor populations in countries that are most vulnerable to such (re-)emerging epidemics [84]. In summary, the current consensus is to focus on (real-time) surveillance rather than prediction.

We here present an overview of these ongoing efforts and conclude with a discussion of initial work on how to assess hypothetical intervention strategies to combat viral spread in a phylogenetic framework.

### 7.1. Formal Assessment of Temporal Signal

An important first step in any phylodynamic analysis is to determine whether the available data contain sufficient temporal signal to estimate the parameters of the molecular clock. Until recently, such assessments relied on visual explorations of the data through regression of root-to-tip genetic distance against sampling time. TempEst [85] is a widely used application for detecting potential issues with data quality (e.g., contamination, recombination or alignment errors), but is not suitable for statistical hypothesis testing and serves rather as a data exploration tool. To formally test whether a heterochronous data set contains sufficient temporal signal, Bayesian Estimation of Temporal Signal (BETS) was developed [86], based on recent advances in Bayesian model selection [87,88]. BETS can determine the statistical support of whether a sufficient amount of molecular evolution has occurred over the sampling time window, which is a key condition—also dubbed the ‘phylodynamic threshold’ [89]—in order to obtain reliable inferences in phylodynamic analyses.

### 7.2. Maximum-Likelihood Phylogeographic Inference

The wide range of studies we discussed in this review point to BEAST as the software package of choice in terms of performing phylogeographic analyses for RABV. However, this does not mean that Bayesian inference is the only possible framework for performing such analyses. Recent years have seen the implementation of discrete phylogeographic models in different software packages that enable maximum-likelihood phylogenetic and phylogeographic inference. Both TreeTime [90] and PastML [91] allow for ancestral location reconstruction based on the same discrete phylogeographic model implemented in BEAST [12]. One important difference between these ML applications and the BEAST implementation lies in the fact that the former both require a phylogeny to be provided as input, with TreeTime taking an unrooted phylogenetic tree as its input and PastML a rooted phylogenetic tree. The fact that multiple applications need to be combined to perform phylogeographic analysis using ML may explain why BEAST is such a widely used software application for performing (joint) phylogeographic analyses. With increasing data set sizes as a result of genomic sequencing efforts, we do expect these ML implementations to become more widely used due to the smaller amount of time they take to obtain a location-annotated ML tree. Recent developments in fact opt for a combination of ML and Bayesian inference approaches in analysing large data sets [92], whereby large data sets are first analysed using ML inference followed by a more in-depth Bayesian phylogeographic analysis of specific clusters of interest. Despite these novel developments, Bayesian phylogeographic inference will remain an important staple of research studies on pathogen spread, owing to its ability to incorporate prior information, the wide range of available models and its inherent capability to take into account sources of uncertainty, to name a few.

### 7.3. Mitigating Sampling Bias

The popularity of the discrete [12] and continuous [13] phylogeographic models has lead to further research into the accuracy and reliability with which these models estimate ancestral locations. In highly important work on structured coalescent models [93], De Maio et al. [65] illustrated the need for more accurate discrete phylogeographic methods in order to overcome the sensitivity to biased sampling. The authors used simulations and empirical analyses to show that their BASTA (BAyesian STructured coalescent Approximation) model does not exhibit sampling-dependent bias and accurately estimates parameter and ancestral reconstruction uncertainty. Despite continuing advances in structured coalescent approaches [94,95], applications of these methods are still confined to rather limited data sets in terms of number of taxa and sampling locations. We refer to Baele et al. [96] for a more in-depth comparison between discrete phylogeography and structured coalescent models. The issue of sampling bias is not the only cause of concern to keep in mind when employing discrete phylogeographic models. Recently, Gascuel and Steel [97] demonstrated that it is generally impossible to accurately estimate both the ancestral locations at the root and the migration rates along the tree branches from the observed data at the tips of the tree. Additionally, and of particular concern, the authors note that the uncertainty of simultaneous estimation is not reduced in a coalescent-based framework when the number of sequences is increased. One possible approach to alleviate these concerns is to include additional metadata (when available) in the reconstruction of the ancestral location states. Lemey et al. [79] developed a framework to integrate individual travel history data in Bayesian discrete phylogeographic inference and showed that this leads to more realistic inferences related to virus spread, in addition to ameliorating the impact of sampling bias by augmenting the phylogeographic analysis with lineages from undersampled or even unsampled locations.

Sampling bias can also be an issue in continuous phylogeographic inference when (a proportion of) samples are missing from certain locations, for example. Kalkauskas et al. [80] have recently proposed to add sequence-free samples from undersampled areas to mitigate this issue, but acknowledge that it may be difficult to assess how many sequences to add from those locations. As an alternative to still perform such inference, the authors propose to use an alternative spatial Λ-Fleming-Viot (ΛFV) process, which is unexplored for use in phylogeographic inference. While this model currently lacks an implementation in popular software applications, Kalkauskas et al. [80] show it to be appropriate to model viral diffusion in cases of endemic spread, with the classic continuous phylogeographic model [13] being more appropriate for recent outbreaks.

Before performing any phylodynamic analysis, data curation efforts may require making difficult decisions in the case of incomplete data, such as imprecise or even missing sampling locations for sequences. In a discrete phylogeography framework [12], ambiguity codes can be employed to specify that sequences were sampled from one of the available locations in the data set. In a continuous phylogeography framework, a polygon defining a uniform prior range of coordinates can be associated with a sequence for which no accurate sampling coordinates are available [76]. A recent extension of this approach allows for incorporation of heterogeneous prior sampling probabilities—of which the sum is constrained to be equal to one—over a geographic area, in the form of a collection of sub-polygons, informed by external data such as previous outbreak locations or host species densities [77].

### 7.4. Assessing (Hypothetical) Intervention Strategies

As a result of a series of devastating epidemics in recent years, such as the 2013–2016 West African Ebola virus outbreak and the ongoing SARS-CoV-2 pandemic, novel phylogenetic and phylodynamic methods have been developed to determine the impact of proposed intervention strategies. Dellicour et al. [98] developed a phylogenetic pruning approach to assess the impact of preventing viral lineage movement over a range of distances on epidemic size and duration, as well as of preventing viral lineage movement to a specific category of administrative areas or to individual administrative areas (based on their population size). Worobey et al. [99] incorporated travel history information into their phylodynamic analyses [79] to uncover that early SARS-CoV-2 introductions into Germany and the west coast of the United States were extinguished by vigorous public health efforts. Dellicour et al. [92] made use of a phylodynamic workflow that balances efficiency with accuracy in order to rapidly analyse a large SARS-CoV-2 data set focused on Belgium, which they used to assess the impact of lockdown measures within the country. Rasigade et al. [100] adapted a previously established phylogenetic method [101] to develop a phylodynamic survival analysis approach to quantify the effect of non-pharmaceutical interventions on the transmission rate of SARS-CoV-2 during the early dissemination phase of the pandemic. We expect these phylogenetic and phylodynamic approaches to play a more important role in years to come, due to their clear aim at limiting viral spread and informing public health in general. 

## Figures and Tables

**Figure 1 viruses-13-01628-f001:**
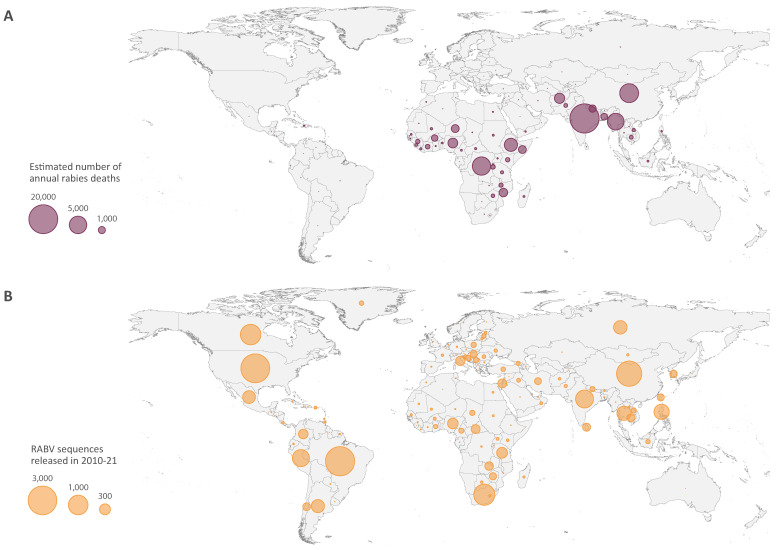
World maps showing the global incidence of the yearly number of deaths due to canine-associated rabies as estimated by Hampson et al. [38] (**A**), and the number of (partial and whole genome) sequences for rabies lyssavirus on GenBank collected between 1 January 1882 and 15 March 2021 (in any host; **B**).

**Figure 2 viruses-13-01628-f002:**
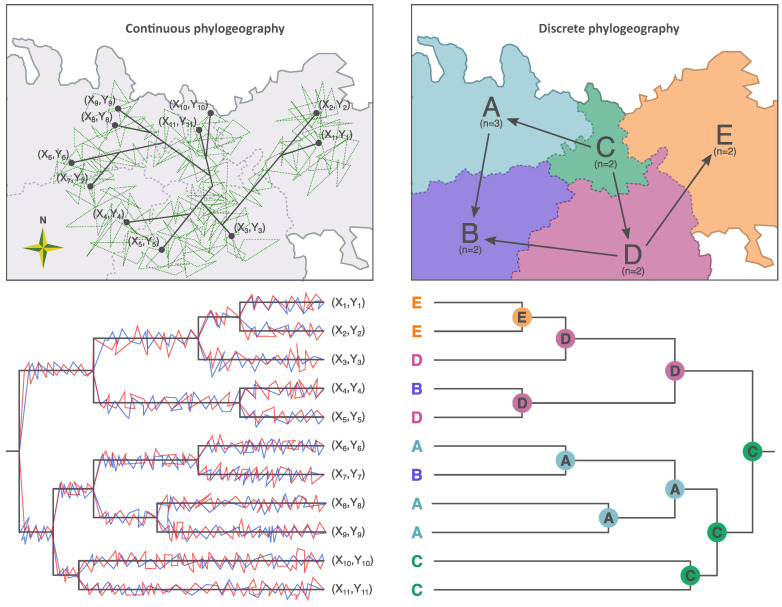
Comparison between continuous and discrete phylogeographic inferences, two methods that have been frequently used to reconstruct the spread of RABV lineages. Both methods can be seen as character mapping approaches: ancestral reconstruction of the longitude X (in blue) and latitude Y (in red) in the case of the continuous phylogeographic inference, and of the discrete location (A, B, C, D, or E) in the case of the discrete phylogeographic inference. The continuous and discrete phylogeographic methods employ a (relaxed) random walk diffusion model (in green) and a discrete diffusion model, respectively. While the former one is spatially explicit and allows the inference of internal nodes across unsampled locations, the latter one requires the preliminary delimitation of discrete locations and does not allow inferring ancestral locations outside this pre-defined set of locations.

**Table 1 viruses-13-01628-t001:** Overview of discrete phylogeographic studies on RABV considered. Many of these studies did not have whole-genome sequences at their disposal. Dogs constitute the species of attention in the majority of studies.

Publication	Year	Sequences	Region	Species
Brunker et al. [41]	2015	59 (whole) + 50 (partial)	Tanzania	dogs
Brunker et al. [6]	2018	152 (whole)	Tanzania	dogs
Omodo et al. [23]	2020	84 (partial)	Uganda	livestock, dogs, jackals and foxes
Hayman et al. [42]	2021	139 + 88 (partial)	Ghana	dogs, cats
Talbi et al. [43]	2010	287 (partial)	North Africa	dogs
Mollentze et al. [44]	2013	636 (partial)	South Africa	dogs
Seetahal et al. [45]	2013	183 (partial)	Trinidad	livestock (bovine, caprine, ovine, equine)
Trewby et al. [46]	2017	289 (whole)	USA-Canada border	raccoon
Streicker et al. [47]	2019	75 (partial)	Central America	vampire bats
Carnieli et al. [48]	2011	71 (partial)	Brazil	dogs
Horton et al. [49]	2015	139 (partial)	Middle East	domestic dogs, wildlife
Dellicour et al. [20]	2019	109 (whole)	Iran	dogs, wolves, jackals, foxes
Tohma et al. [50]	2014	233 (partial)	Philippines	dogs
Dibia et al. [51]	2015	63 (partial)	Indonesia	dogs, cattle, goat, cat
Lin et al. [52]	2016	220 (partial)	Taiwan	ferret badgers
Lan et al. [53]	2017	156 (partial)	Taiwan	ferret badgers
Yu et al. [54]	2012	110 + 90 (partial)	China	dogs, cats, deer, raccoon dogs, striped field mice, ferret badgers
Guo et al. [55]	2013	232 (partial)	China	dogs
Ma et al. [37]	2017	36 (partial)	Shaanxi (China)	dogs
Zhang et al. [56]	2017	452 (partial)	Yunnan (China)	dogs, humans
Tian et al. [19]	2018	1034 (partial)	Yunnan (China)	dogs
Wang et al. [57]	2019	112 (partial)	China and neighbours	dogs
Yu et al. [58]	2021	155 (partial)	North and South Asia	dogs, red fox, swift fox, raccoon dogs, cow, sheep camel

**Table 2 viruses-13-01628-t002:** Overview of continuous phylogeographic studies on RABV considered. As was the case for the discrete phylogeographic studies, many analyses did not have whole-genome sequences at their disposal.

Publication	Year	Taxa (Whole Genomes)	Region	Species
Brunker et al. [6]	2018	152 (whole)	Tanzania	dogs
Omodo et al. [23]	2020	84 (partial)	Uganda	livestock, dogs, jackals and foxes
Kuzmina et al. [22]	2013	241 (partial)	North America	skunks
Pepin et al. [67]	2017	73 (partial)	Colorado (USA)	skunks
Musial et al. [68]	2018	193 (partial)	Florida (US)	raccoons
Carnieli et al. [69]	2013	53 (partial)	Brazil	dogs
Vieira et al. [70]	2013	41 (partial)	Brazil	cattle, vampire bat
Torres et al. [2]	2014	790 + 547 (partial)	Argentina	vampire bat
Streicker et al. [71]	2016	264 (partial)	Peru	vampire bats
Streicker et al. [47]	2019	75 + 40 (partial)	Costa Rica	vampire bats
Fusaro et al. [72]	2013	160 (partial)	Italy and the Balkans	foxes
Tian et al. [19]	2018	1034 (partial)	Yunnan (China)	dogs
Dellicour et al. [20]	2019	109 (whole)	Iran	dogs, wolves, jackals, foxes
Dellicour et al. [5] (meta-analysis)	2017	[2,10,22,43,70]	North America, North Africa, Eastern Argentina, Eastern Brazil	skunk, raccoon, domestic dog, vampire bats

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
