# Peer review of "On the Use of Phylogeographic Inference to Infer the Dispersal History of Rabies Virus: A Review Study"

_viruses, 2021, doi:10.3390/v13081628_

Round 1

Reviewer 1 Report

The authors present a well written, well-thought out review describing how genotypes of RABV virus gave rise to certain RABV strains in geographical areas.  They also describe how the various hosts and their geographic distribution play a role in RABV spread.  The authors present a meta-data analysis based on genomics and other published data.  They describe a history of rabies, but could spend a few more paragraphs on how the methodology could be used to predict future outbreaks or applications to other zoonotic viruses.

Author Response

We thank the Reviewer for the positive evaluation of our work. We have incorporated the Reviewer’s comments by including additional paragraphs in our revised manuscript. 

On the topic of applying the methods discussed to other zoonotic viruses, we have added the following paragraph in the section “Recommendations and useful resources”: “While we discuss the application of phylogeographic methods on rabies virus (RABV) datasets here, it should be noted that these methods could also be used on genomic data from other zoonotic viruses, provided the data contain sufficient temporal signal. We elaborate more about the formal assessment of a temporal signal on available data in the next section.” 

Predicting future outbreaks is however a controversial topic, but we understand the Reviewer’s request to touch upon this. As such, we have opted to caution against attempts at predicting future outbreaks in favour of investing in surveillance, and have added the following paragraph in the section “Novel methodological developments and future perspectives”: “While it is tempting to try to leverage these phylogeographic methods to predict (the locations of) future outbreaks through some form of forward simulation, this is far from straightforward as it entails drawing reliable patterns from relatively rare past events (Geoghegan & Holmes, 2017). Additionally, the circumstances surrounding each outbreak are very dynamic, complicated and unique due to interactions between virus genetics, ecology and host factors (Grubaugh et al., 2019). This makes it extremely difficult to build a predictive model because such a model must be ‘trained’ on a very wide range of realistic scenarios in order to be reasonably accurate. Given such difficulties, it has been suggested that the most effective and realistic way to fight outbreaks is to monitor populations in countries that are most vulnerable to such (re-)emerging epidemics (Holmes et al., 2018). In summary, the current consensus is to focus on (real-time) surveillance rather than prediction.”

References:

  1. Geoghegan, Jemma L., and Edward C. Holmes. "Predicting virus emergence amid evolutionary noise." 2017. Open Biology 7(10): 170189.
  2. Grubaugh, Nathan D., et al. "Tracking virus outbreaks in the twenty-first century." 2019. Nature Microbiology 4(1), 10-19.
  3. Holmes, Edward C., Andrew Rambaut, and Kristian G. Andersen. "Pandemics: spend on surveillance, not prediction." 2018. Nature 558, 180-182.

Reviewer 2 Report

Nahata et al., On the use of phylogeographic inference to infer the dispersal history and dynamics of rabies virus lineages: a review study.

This review is well-written and provides informative data by bringing together numerous published studies using available sequence data with a focus on environmental factors relevant to the spread of rabies virus. The review concludes with an overview of recent methodological  developments in phylogeographic and phylodynamic analyses of rabies virus. Overall, I do not have any serious concerns. The title however is verbose and could be shortened for clarity and preciseness. My principal unease with this review is that the analysis was undertaken on a number of partial and whole genome sequences for rabies lyssavirus collected between 2010 and 2021. For this reason, a large number of pre-2010 partial and whole genome sequences for rabies lyssavirus from unique rabies-endemic regions of the world have not been included. In order to support a straightforward and simplified approach, I suggest that a supplementary file of pre-2010 rabies lyssavirus sequences as a table is included with appropriate citations, species and region(s).

Minor Comments:  

Again, whilst it is appreciated that a complete review of all manuscripts between 2010 and 2020 reporting rabies lyssavirus is challenging, the authors’ may also wish to consider the following:

  1. Johnson, et al., 2010. Rabies epidemiology and control in Turkey: past and present. Epidemiology and Infection 138(3); 305-12.
  2. Turcitu et al., 2010. Molecular epidemiology of rabies virus in Romania provides evidence for a high degree of heterogeneity and virus diversity. Virus Research 150(1-2); 28-33.
  3. Zieger, et al., 2014. The phylogeography of rabies in Grenada, West Indies, and implications for control. PLoS Neglected Tropical Diseases 8(10):e3251.
  4. Fischer, et al., 2018. Defining objective clusters for rabies virus sequences using affinity propagation clustering. PLoS Neglected Tropical Diseases 12(1); e0006182.

Author Response

We thank the Reviewer for these suggestions and for the appreciation of our work. Incorporating the Reviewer’s comment regarding the title of our previous manuscript, we have now shortened it in this revision to: “On the use of phylogeographic inference to infer the dispersal history of rabies virus: a review study”. 

Our main goal for this review manuscript was to highlight studies that use the phylogeographic methods (discrete and continuous phylogeography, introduced in 2009 and 2010 respectively) described in the manuscript and how these methods have helped infer the dispersal history and dynamics of RABV lineages. Concerning the Reviewer’s comment on the exclusion of genomic data and studies pre-2010, we realise that while this is indeed unfortunate, our aim was to review the use of the phylogeographic methodologies mentioned above since their development (Lemey et al., 2009; Lemey et al., 2010), and not to provide an overview of all available RABV sequences. As noted by the Reviewer, unfortunately not all rabies-endemic regions have been studied using these methods, and thus some important sequences may have been excluded as a result. While the tables provide a useful oversight of some of the sequence data available, this is hence not an exhaustive overview, nor was it meant as such. We apologize for the confusion, and have now added the following text in the “Introduction” section of our revised manuscript: “Some important rabies-endemic regions (most notably India) were not (yet) studied using these phylogeographic inference methods, and the studies presented in this review hence do not offer a complete overview of the dispersal history of RABV lineages across the world”. Furthermore, we no longer mention the studies Biek et al. (2007) and McElhinney et al. (2011)  in Table 1/1A as well as the section “Discrete phylogeographic inference”. In the previous version of our manuscript, we had included these studies as they represent important work in the field of RABV phylogeography but they have now been removed as these studies do not utilize the aforementioned methodologies.

However, we agree that a more comprehensive look at the spread of rabies is useful. The second subfigure in Figure 1 does in fact show an overview of all available RABV sequences, regardless of which method has been used to study them. We had made a mistake in the caption of this figure in the previous version of our manuscript and this figure does in fact include all RABV sequences available on GenBank, collected between 01-01-1882 and 15-03-2021. We have changed the caption in question in the revised manuscript to more accurately reflect this.  

Finally, we thank the Reviewer for suggesting additional literature for our consideration. We have now cited the study - Zieger et al. (2014) in the sentence from the previous version of our manuscript in the section “(Lack of) genomic data, geographic scope and methodological expertise”: “In our literature review, we encountered a number of epidemiological studies in which the sequence data were accompanied by either precise geographic coordinates or discrete locations, but no phylogeographic methodologies were applied”. Respectfully, we deem the other studies mentioned to be outside the scope of our current review study.

References:

  1. Biek, Roman, et al. "A high-resolution genetic signature of demographic and spatial expansion in epizootic rabies virus." Proceedings of the National Academy of Sciences 104.19 (2007): 7993-7998.
  2. Lemey, Philippe, et al. "Bayesian phylogeography finds its roots." PLoS computational biology 5.9 (2009): e1000520.
  3. Lemey, Philippe, et al. "Phylogeography takes a relaxed random walk in continuous space and time." Molecular biology and evolution 27.8 (2010): 1877-1885.
  4. McElhinney, L. M., et al. "Molecular diversity and evolutionary history of rabies virus strains circulating in the Balkans." Journal of General Virology 92.9 (2011): 2171-2180.
  5. Zieger, et al., 2014. The phylogeography of rabies in Grenada, West Indies, and implications for control. PLoS Neglected Tropical Diseases 8(10):e3251.